# A novel probe to assess cytosolic entry of exogenous proteins

Qiao Lu[1], Jeff E. Grotzke[1] & Peter Cresswell[1]

Dendritic cells use a specialized pathway called cross-presentation to activate CD8$^+$ T cells by presenting peptides from exogenous protein antigens on major histocompatibility complex class I molecules. Considerable evidence suggests that internalized antigens cross endocytic membranes to access cytosolic proteasomes for processing. The mechanism of protein dislocation represents a major unsolved problem. Here we describe the development of a sensitive reporter substrate, an N-glycosylated variant of Renilla luciferase fused to the Fc region of human IgG1. The luciferase variant is designed to be enzymatically inactive when glycosylated, but active after the asparagine to aspartic acid conversion that occurs upon deglycosylation by the cytosolic enzyme N-glycanase-1. The generation of cytosolic luminescence depends on internalization, deglycosylation, the cytosolic AAA-ATPase VCP/p97, and the cytosolic chaperone HSP90. By incorporating a T cell epitope into the fusion protein, we demonstrate that antigen dislocation into the cytosol is the rate limiting step in cross-presentation.

---

[1] Department of Immunobiology, Yale University School of Medicine, New Haven, CT 06520, USA. Correspondence and requests for materials should be addressed to P.C. (email: peter.cresswell@yale.edu)

Certain toxins that inhibit protein translation, such as ricin and diphtheria toxin, access the cytosol of cells following endocytosis. The amount of toxin that enters the cytosol is difficult to measure, but is generally considered to be small[1,2]. External growth factors can also be transferred into the nucleus of fibroblasts where they act as transcription factors[3]. In addition, cell-penetrating peptides can transport associated proteins across tissue and cell membranes and gain access to the cytosol[4]. Immunological studies have uncovered a broader role for the cytosolic entry of external proteins in the immunological phenomenon of cross-presentation. Here protein antigens acquired by endocytosis or phagocytosis are translocated across the endosomal/phagosomal membrane and degraded by cytosolic proteasomes. The resulting peptides are translocated by transporter associated with antigen processing (TAP) into the endoplasmic reticulum (ER) or back into the endosome/phagosome where they can bind to major histocompatibility complex class I (MHC-I) molecules. These MHC-I-peptide complexes then traffic to the cell surface for presentation to CD8$^+$ T cells. The primary cell types that mediate cross-presentation in vivo are specific subsets of dendritic cells (DCs), and the process is essential for the initiation of cytotoxic T cell responses and for maintaining immune tolerance[5,6].

The underlying mechanism of antigen transfer to the cytosol is poorly understood. It has been suggested that the ER-associated degradation (ERAD) machinery, which translocates misfolded proteins from the ER into the cytosol, is involved. ER components can be recruited to phagosomes, including components of the peptide loading complex that normally facilitate MHC-I peptide binding in the ER, namely tapasin, TAP, ERp57, and calreticulin. Recruitment involves the fusion by a Sec22b-dependent mechanism of vesicles derived from the ER-Golgi intermediate compartment with the phagosomal membrane[7–14]. It has been suggested that Sec22b may not be important[15], but its requirement for in vivo cross-presentation has been confirmed using Sec22b knockout mice[16]. Sec61, postulated to be a translocon involved in ERAD, has also been implicated[17], although recent data has cast doubt on its role in both ERAD and cross-presentation[18]. The AAA ATPase VCP/p97, known to be required for ERAD, also appears to be important for cross-presentation, perhaps in both cases by extracting proteins from a dedicated channel[11,19]. The delivery of internalized toxins into the cytosol may require ERAD components[2], but using siRNA approaches we were unable to show that major defined ERAD channel components, such as Hrd1, gp78, HERP, and Derlin-1, are involved in cross-presentation[20]. It is conceivable that no precise channel is involved: recently it has been suggested that antigens are released into the cytosol by endosomal "leakage" caused by lipid peroxidation induced by reactive oxygen species produced by the NADPH oxidase NOX2[21].

Tools that allow direct measurement of protein dislocation into the cytosol are highly desirable. T cell detection of the endpoint of the process, i.e., surface MHC-I-peptide complexes, is sensitive and straightforward but neither quantitative nor specific for the dislocation step. The addition to intact cells of cytochrome C can induce apoptosis by cytosolic caspase activation, but this is not quantitative and requires high concentrations of protein[21,22]. Another approach uses the bacterial enzyme β-lactamase, but this requires pre-loading the cells with a cytosolic fluorescent substrate[12,21]. Here, we describe a novel derivative of Renilla luciferase (RLuc), an enzyme that produces bioluminescence as a product of substrate catalysis. We describe an inactive glycosylated variant that is activated when the enzyme enters the cytosol. The restoration of activity for this deglycosylation-dependent variant (ddRLuc) relies on the asparagine (N) to aspartic acid (D) conversion that occurs when the glycan is removed by the cytosolic enzyme N-glycanase-1 (NGLY1), the product of the gene Ngly-1[23–25]. Fusion of ddRLuc with the Fc region of human IgG1 (ddRLuc-Fc) stabilizes the enzyme, facilitates purification, and allows binding and internalization by cell surface Fc receptors (FcR). We show that a variety of cells, including human and mouse DCs, generate luminescence following ddRLuc-Fc internalization, indicating successful cytosolic dislocation of the substrate. A variant incorporating a well-defined epitope derived from ovalbumin (OVA) allowed us to directly correlate cytosolic access with cross-presentation.

## Results

**Generation and characterization of ddRLuc-Fc**. We previously developed deglycosylation-dependent fluorescent proteins to study retro-translocation during ERAD[26]. Starting with RLuc that contains a C124A mutation (hereafter called WT RLuc) for protein stabilization and enhanced bioluminescence[27], we initially mutated nine surface aspartic acid (D) or glutamic acid (E) residues to generate a deglycosylation-dependent derivative of RLuc. These were individually mutated to asparagine (N), and simultaneously the second amino acid downstream was mutated to serine (S) or threonine (T) to generate a glycosylation motif (the N version). A second set of mutants combined the D residues with downstream S or T residues, to generate the expected sequence following deglycosylation (the D version). The N and D versions were cytosolically expressed in 293T cells and total cell lysates assessed for luciferase activity. Three variants showed the desired pattern: almost complete lack of activity for the N version but activity for the D version, with substitution at position 290 leading to a 1000-fold difference in luminescence generation (Supplementary Fig. 1). To assess deglycosylation-dependence, we added a signal sequence to the N versions (SS-N) to allow N-glycosylation in the ER. Cells transfected with SS-N variants were treated with epoxomicin (Epox), a potent and selective proteasome inhibitor[28], to allow the accumulation of active RLuc, retrotranslocated via the ERAD pathway, in the cytosol. The cell-permeable NGLY1 inhibitor z-VAD(OMe)-FMK (zVAD) was used to confirm that any gain of luciferase activity was dependent on deglycosylation[29]. Of the tested mutants, only SS-N290 (hereafter called ddRLuc) displayed vigorous deglycosylation-dependent activity. This activity was extremely sensitive to proteasomal degradation (~60-fold increase in relative light units (RLU) after Epox treatment) and required NGLY1-mediated deglycosylation (~20-fold decrease in RLU after co-treatment with Epox and zVAD), making it a highly sensitive translocation reporter (Supplementary Fig. 2).

ddRLuc was fused to human IgG1 Fc to facilitate purification (Fig. 1a). ddRLuc-Fc retained deglycosylation-dependent activity when expressed in 293T cells and used as an ERAD substrate (Fig. 1b, c). It was isolated from cell extracts using protein A-Sepharose and purity was confirmed by Coomassie Blue staining (Fig. 1d, left panel) and western blot (Fig. 1d, right panel). ddRLuc-Fc was predominantly glycosylated (Fig. 1d, red arrow) and mainly purified as a dimer (Fig. 1d, yellow arrow), which allows high-affinity Fc receptor binding[30]. The bacterial enzyme endo-β-acetylglucosaminidase H (Endo H) efficiently removed the glycan, as did the bacterial recombinant Peptide:N-glycanase F (PNGase F) (Fig. 1e), revealing that the introduced N-glycan is mainly in the high mannose form[31,32]. However, unlike PNGase F, which has the same cleavage specificity as NGLY1[33], Endo H does not mediate the D to N conversion[31] and consequently failed to generate luciferase activity (Fig. 1f). Glycosylated ddRLuc-Fc has a short half-life in PBS at 37 °C ($t_{1/2}$ ~ 20 min), but deglycosylation by PNGase F stabilizes the protein, extending the half-life by a factor of five ($t_{1/2}$ ~ 100 min) (Fig. 1g).

**Validation of ddRLuc-Fc dislocation in cell lines**. We initially analyzed dislocation using an engineered 293T cell line co-expressing human FcRγIIA and mouse MHC-I K[b] that can internalize and cross-present exogenous antigen[10]. For endocytosis experiments, purified ddRLuc-Fc was added directly to 293T-FcRγIIA-K[b] prior to incubation at 37 °C, while for phagocytosis experiments it was first passively adsorbed to 3 μm latex beads. The proteasome and NGLY1 inhibitors described above were used to confirm that any luminescence generated was enhanced by proteasome inhibition and dependent on NGLY1-mediated deglycosylation after both phagocytosis and endocytosis (Fig. 2a, b). The cytosolic chaperone Hsp90[34,35] and VCP/ p97[11,19] have been shown to facilitate protein dislocation into the cytosol, and, consistent with this, addition of the Hsp90 inhibitor radicicol (Rad) or the VCP/p97 inhibitor CB5083 reduced activity (Fig. 2a, b). ddRLuc-Fc activation was inhibited by a combination of cytochalasin D (Cyto D) and Dynasore (Dyn), which block phagocytosis and endocytosis, respectively (Fig. 2a, b), ruling out the possibility that extracellular ddRLuc-Fc could be activated by NGLY1 released from dead cells. The vigorous activity generated after phagocytosis (Table 1) allowed us to directly visualize the signal in intact cells by luminescence microscopy. Epox-treated cells produced strong luminescence, while DMSO control samples emitted a weak signal, and co-treatment with Epox and

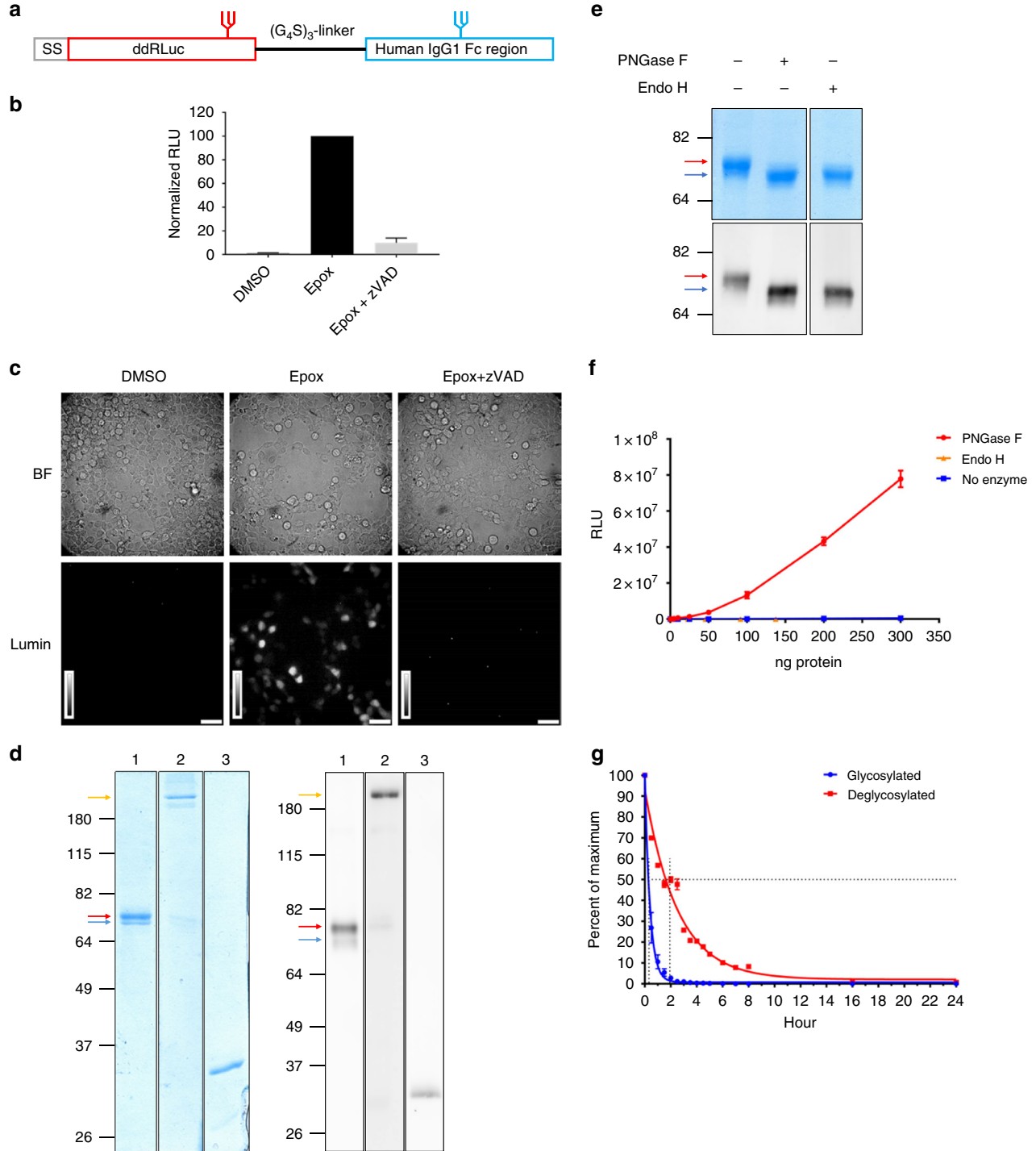

zVAD completely suppressed the signal (Fig. 2c), confirming its deglycosylation dependence. To confirm the cytosolic location of the activity, we used Streptolysin O (SLO) to permeabilize 293T-FcγIIA-K$^b$ cells after phagocytosis and isolated the released cytosolic material. Western blots showed that purified cytosol was negative for ER, endosomal, or lysosomal/phagosomal proteins (Grp94, prosaposin, and cathepsin D), but contained the cytosolic enzyme glyceraldehyde 3-phosphate dehydrogenase (GAPDH) (Supplementary Fig. 3a). The cytosol released from Epox-treated cells contained luciferase activity, and the effects of the various inhibitors were similar to those seen using total cell lysates (compare Supplementary Fig. 3b and Fig. 2a). Of note, PNGase F digestion of the cytosol demonstrates a population of dislocated but still glycosylated ddRLuc in Epox + zVAD treated cells, demonstrating that dislocation precedes deglycosylation (compare Supplementary Fig. 3b and c). In addition, we confirmed that most (approximately 70%) of the dislocated ddRLuc present in the cytosol from Epox-treated cells was already deglycosylated based on lack of binding to Concanavalin A-Sepharose (Supplementary Fig. 3d, e). When the cells were incubated in the presence of zVAD prior to isolation of the cytosol, this fraction was reduced to less than 10%, as predicted if deglycosylation mediated by NGLY1 occurs in the cytosol.

To reveal the total intracellular ddRLuc-Fc, i.e., the combination of deglycosylated and residual glycosylated RLuc, cell lysates were treated with PNGase F. As expected, because of proteasomal degradation, the DMSO control samples contained less total ddRLuc-Fc than the Epox-treated samples, and the addition of Cyto D and Dyn significantly reduced total intracellular activity (Supplementary Figs. 3c and 4). However, the reduction in activity observed when zVAD was combined with Epox was not observed in the PNGase F-digested samples (compare Fig. 2 and Supplementary Fig. 4), indicating that zVAD does not affect ddRLuc-Fc internalization or dislocation. We did observe a reduction in total ddRLuc-Fc after co-treatment with Epox and CB5083 (Supplementary Figs. 3c and 4). This strongly supports a role for VCP/p97 in facilitating dislocation prior to deglycosylation by NGLY1, and also suggests that preventing dislocation results in degradation within the endocytic pathway. The similar effect of combining Epox and Rad (Supplementary Figs. 3c and 4) could be because Hsp90 inhibition affects dislocation[34] or, as previously suggested[35], prevents refolding of ddRLuc-Fc in the cytosol.

ddRLuc-Fc activity was also detected in other cell lines where it might not be expected, including TRal, an Epstein-Barr virus-transformed B cell line that expresses an uncharacterized FcR[36], which is functional for endocytosis but not for phagocytosis (Supplementary Fig. 5a–d). Surprisingly, FcR-negative cells, such

as 293T cells (Supplementary Fig. 5e–h) and Expi293F cells (Supplementary Fig. 5i–l), displayed proteasome-sensitive, deglycosylation-dependent activity, although this was weaker than 293T-FcγIIA-K$^b$ cells (Table 1), presumably because of the absence of an FcR. Thus ddRLuc-Fc can be used to measure cytosolic dislocation independently of Fc receptor expression.

**Kinetics of ddRLuc-Fc dislocation.** The kinetics of luciferase activation was examined over 24 h following internalization by 293T-FcγIIA-K$^b$ cells. For endocytosis experiments, purified ddRLuc-Fc was allowed to bind to 293T-FcγIIA-K$^b$ at 4 °C and the cells were washed prior to incubation at 37 °C to initiate internalization. For phagocytosis, ddRLuc-Fc-coated beads were added at 4 °C to cells that were then washed and warmed to 37 °C. Epox and other inhibitors were added to assess their effects on the generation of luciferase activity, leading to the concern that cell viability might be affected over such a long time course. We found no effects on viability of any drug individually or in combination up to 8 h, but more prolonged incubation with a mixture of Epox and Rad did result in a loss in viability (Supplementary Fig. 6a, b).

In the presence of Epox, activity that was predominantly deglycosylation-dependent was detected as early as 2 h, and throughout the time course it was dependent on Hsp90 (Fig. 3). Luciferase activity continued to accumulate up to 12 h following phagocytosis (Fig. 3a) and up to 9 h following endocytosis (Fig. 3b). In the case of phagocytosis, a significantly higher proportion of the luciferase activity was independent of proteasome inhibition but did require NGLY1 activity (Fig. 3a), perhaps reflecting proteasome saturation because of greater substrate entry into the cytosol. By treating aliquots of the samples from Epox-treated cells at each time point with PNGase F, to account for properly folded but still glycosylated ddRLuc-Fc, we showed that the fraction of deglycosylated ddRLuc-Fc grew over time, rising steadily until it plateaued around the 8–10 h time-point in the context of phagocytosis (Supplementary Fig. 6c) or around the 6–8 h time-point in the context of endocytosis (Supplementary Fig. 6d). This indicates that at this stage deglycosylation was complete and the vast majority of the activity was in the cytosol. Using these data, we could estimate the stability of the enzyme following dislocation, obtaining a $t_{1/2}$ of 10–12 h, which is much longer than that of the deglycosylated enzyme in solution (~100 min, Fig. 1g). The high protein content of the cytosol plus the presence of cytosolic chaperones may combine to stabilize the protein. To examine this question, we added the Hsp90 and VCP/97 inhibitors to the cells 15 h after phagocytosis, past the peak of the luciferase activity, and assessed

**Fig. 1** Generation and characterization of ddRLuc-Fc. **a** Schematic description of ddRLuc-Fc. The red glycan at N290 within ddRLuc is required for the deglycosylation-dependent activity. The blue glycan in the CH2 region of human IgG1 Fc is required for efficient binding to Fc receptor (FcR)[50]. **b**, **c** ddRLuc-Fc shows proteasome inhibition- and NGLY1 deglycosylation-dependent activity. ddRLuc-Fc-transfected 293T cells were incubated with DMSO alone, 200 nM Epox in DMSO, or a combination of 200 nM Epox and 20 μm zVAD in DMSO, at 37 °C for 6 h. In **b** equal amounts of cell lysates were tested for luciferase activity as described in Methods; activities are normalized to that seen with Epox only, which is set to 100 (bars represent the mean +/−s.d. of one representative experiment with triplicates). In **c** intact cells were examined by luminescence microscopy (BF: bright field; Lumin: luminescence; scale bar = 50 μm; brightness bar inserts: 0 to 255, linear scale). **d** Purification of recombinant ddRLuc-Fc. Coomassie Blue staining (left panel) and western blot (right panel) of purified ddRLuc-Fc. Lane 1: purified ddRLuc-Fc in reducing sample buffer; Lane 2: purified ddRLuc-Fc in non-reducing sample buffer; Lane 3: recombinant RLuc in reducing sample buffer. Red arrow indicates glycosylated ddRLuc-Fc; blue arrow indicates unglycosylated ddRLuc-Fc; yellow arrow indicates ddRLuc-Fc dimer. **e**, **f** In vitro deglycosylation by PNGase F or Endo H followed by SDS gel electrophoresis (**e**, upper panel: Coomassie Blue staining; lower panel: western blot) or luciferase assays (**f**, points represent the mean +/−s.d. of one representative experiment with triplicates). **g** Relative stabilities of glycosylated (blue line) and deglycosylated (red line) ddRLuc-Fc. Samples were incubated at 37 °C for various times and immediately examined by luciferase assays. The horizontal dashed line refers to 50% activity compared to initial activity, and the vertical dash line indicates the $t_{1/2}$. Points represent the mean +/− s.d. of one representative experiment with triplicates. In **b–g**, representative data from three independent experiments are shown

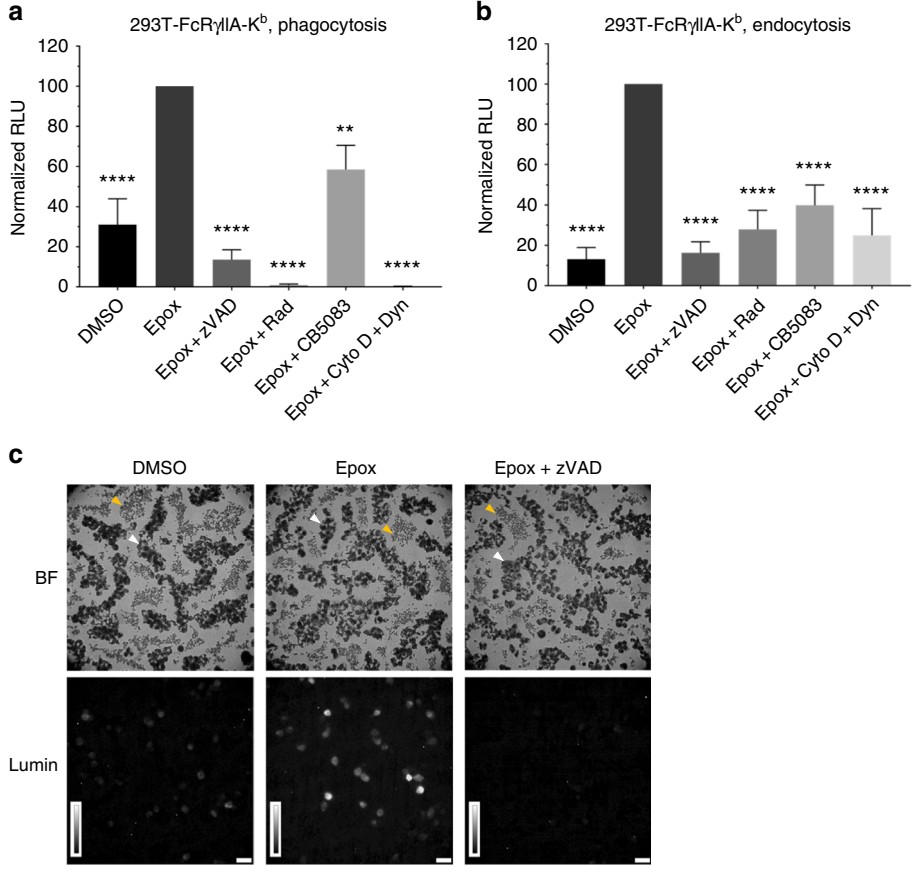

**Fig. 2** ddRLuc-Fc dislocation in 239T-FcRγIIA-K$^b$ cells. **a, b** The 293T-FcRγIIA-K$^b$ cells were incubated with ddRLuc-Fc-bound 3 μm latex beads for phagocytosis (**a**) or 100 μg mL$^{-1}$ soluble ddRLuc-Fc for endocytosis (**b**) at 37 °C for 6 h. Cells were washed, lysed, and the lysates subjected to luciferase assays immediately after lysis to quantify the amount of active ddRLuc-Fc. RLU on the y axis are normalized to the activity obtained with Epox alone (set to 100), and all inhibitors were dissolved in DMSO. Drugs were used at the following concentrations: 200 nM Epox, 20 μm zVAD, 32 μm Rad, 1 μm CB5083, 2.5 μg mL$^{-1}$ Cyto D plus 100 μm Dyn. Bars represent the mean +/−s.d. of at least four independent experiments per treatment (paired two-tailed t-test, **p < 0.01; ****p < 0.0001). **c** Intact 293T-FcRγIIA-K$^b$ cells were incubated with ddRLuc-Fc-bound 3 μm latex beads for phagocytosis and analyzed by luminescence microscopy 9 h post feeding (BF: bright field; Lumin: luminescence; scale bar = 50 μm; brightness bar inserts: 0 to 255, linear scale). White arrow head marks cells; yellow arrow head marks 3 μm latex beads that were not internalized. Representative data of three independent experiments are shown

**Table 1 Representative RLU of DMSO and Epox-treated samples from different cell types**

| RLU[a] | Phagocytosis | | Endocytosis | |
|---|---|---|---|---|
| | DMSO (x10$^4$) | Epox (x10$^4$) | DMSO (x10$^4$) | Epox (x10$^4$) |
| 293T-FcRγIIA-K$^{b\,b}$ | 14.57 | 69.49 | 0.27 | 2.16 |
| TRal[c] | 0.24 | 0.33 | 0.06 | 1.01 |
| 293T[d] | 0.07 | 0.14 | 0.04 | 0.19 |
| Expi293F[e] | 0.11 | 0.48 | 0.56 | 1.90 |
| hDC[f] | 0.23 | 2.24 | 0.11 | 1.32 |
| BMDC[g] | 0.75 | 8.82 | 22.59 | 24.03 |
| BMDC[h] | 0.22 | 9.06 | 18.69 | 18.29 |

[a]RLU was measured directly from 10 μL total cell lysate. Mean value of background-subtracted normalized RLU from a representative experiment is shown
[b]2 × 10$^5$ 293T-FcRγIIA-K$^b$ cells were assayed 6 h after feeding, and Epox was used at 200 nM for proteasome inhibition
[c]2 × 10$^5$ TRal cells were assayed 6 h after feeding, and Epox was used at 800 nM
[d]2 × 10$^5$ 293T cells were assayed 6 h after feeding, and Epox was used at 200 nM
[e]2 × 10$^5$ Expi293F cells were assayed 6 h after feeding, and Epox was used at 200 nM
[f]2 × 10$^5$ hDCs were assayed 6 h after feeding, and Epox was used at 800 nM
[g]5 × 10$^5$ BMDCs were assayed 3 h after feeding, and Epox was used at 800 nM
[h]5 × 10$^5$ BMDCs were assayed 3 h after being fed ddRLuc-Fc$^{CHO−}$, and Epox was used at 800 nM

the effect of this on the rate of decay. We found that inhibition of Hsp90 accelerated the loss of luciferase activity, reducing the $t_{1/2}$ of the dislocated deglycosylated enzyme to approximately 2 h. Thus, Hsp90 appears to play a role in stabilizing the cytosolic enzyme (Supplementary Fig. 7). Inhibition of VCP/p97, which is involved in the dislocation event, had no effect, consistent with the activity already being in the cytosol at the time of addition of the inhibitor.

**ddRLuc-Fc dislocation in DCs.** Human monocyte-derived DCs (hDCs) were tested for the generation of luciferase activity 6 h after ddRLuc-Fc binding to the cell surface. Similar to 293T-FcRγIIA-K$^b$ cells, we observed deglycosylation-dependent ddRLuc-Fc activity after both phagocytosis and endocytosis. Again, the signal was dependent on proteasome inhibition as well as active Hsp90 and VCP/p97, and required internalization (Fig. 4a, b).

We performed similar experiments using mouse bone marrow-derived DCs (BMDCs), but using 3 h as a selected time point because these cells tended be activated during prolonged incubation, known to result in reduction in internalization and cross-presentation[37,38]. Dislocation after phagocytosis was similar to that seen with hDCs and 293T-FcRγIIA-K$^b$ cells (Fig. 4c). However, the results for endocytosis were quite different in that considerable luminescence signal was detected that was not dependent on

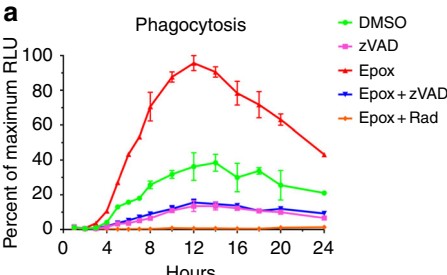
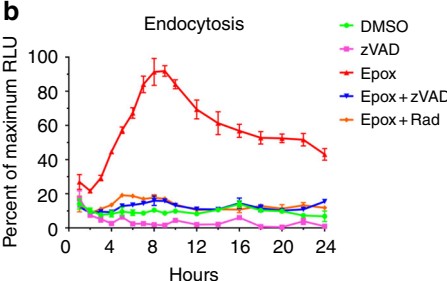

**Fig. 3** Dislocation kinetics of ddRLuc-Fc in 239T-FcγIIA-K$^b$ cells. 293T-FcγIIA-K$^b$ cells were incubated with ddRLuc-Fc-bound 3 µm latex beads for phagocytosis (**a**) or 100 µg mL$^{-1}$ soluble ddRLuc-Fc for endocytosis (**b**) at 37 °C in the presence or absence of various drugs (200 nM Epox, 20 µm zVAD, or 32 µm Rad). Cell lysates were prepared and analyzed as described in Fig. 2 at each time point, and RLU is represented as the percent of maximum value. Points represent the mean +/−s.d. of one representative experiment with duplicates. Representative data of three independent experiments are shown

proteasome inhibition, NGLY1-mediated deglycosylation, Hsp90 activity or VCP/p97 activity (Fig. 4d). It is likely that this correlates with residual activity of glycosylated ddRLuc-Fc (Table 1 and Supplementary Table 2), and reflects enhanced uptake combined with either reduced degradation within the endocytic pathway or greater cytosolic accumulation without deglycosylation. The high activity revealed upon PNGase F treatment (Supplementary Table 1) is consistent with either of these scenarios, although the lack of an effect of Hsp90 and VCP/p97 inhibition (compare Supplementary Fig. 8a–c and d) suggests that prolonged residence of ddRLuc-Fc in the endosome is more likely.

Many investigators have shown that BMDCs cross-present soluble OVA, although this may be influenced by the time of exposure to antigen and their state of maturation[37]. However, the results in Fig. 4d suggested that endocytosed ddRLuc-Fc did not access the cytosol in BMDCs. To address whether lack of cytosolic entry after endocytosis results in poor cross-presentation, we inserted two copies of the H2-K$^b$-restricted OVA-derived epitope SIINFEKL into the linker region of ddRLuc-Fc, generating ddRLuc-Fc$^{OVA}$, a substrate capable of simultaneously monitoring dislocation and cross-presentation (Fig. 5a). When soluble ddRLuc-Fc$^{OVA}$ was used with BMDCs, neither proteasome-inhibitor-dependent luciferase activity nor cross-presentation were observed (Fig. 4e, f, right). However, both proteasome inhibitor-dependent luciferase activity and cross-presentation were readily detected following phagocytosis of beads coated with ddRLuc-Fc$^{OVA}$ (Fig. 4e, f, left). In the endocytosis experiments ddRLuc-Fc$^{OVA}$ was added at only 50 µg mL$^{-1}$. Data consistent with these data but using soluble OVA also found no cross-presentation after a short incubation[37]. Conceivably, small aggregates in the high concentrations of OVA often used with BMDCs may be internalized by phagocytosis rather than classical endocytosis. Successful cross-presentation with lower concentrations of OVA[39] may be a consequence of induction of BMDC maturation over the time course of the experiments, although fully addressing this issue will require further experimentation.

Overall these results point to substantial differences in the capabilities of different DC types and/or DCs isolated from different species to drive productive antigen dislocation during cross-presentation. This may have significant implications for adaptive immunity.

**Antigen dislocation and cross-presentation are correlated**. To examine the relationship between cytosolic dislocation and cross-presentation, ddRLuc-Fc$^{OVA}$ was passively bound to 3 µm beads and added to 293T-FcγIIA-K$^b$ cells. Cells were harvested at 8 h and used to measure dislocation using the luciferase assay and cross-presentation using a T cell assay. Epox treatment enhanced

luminescence by fivefold over the control while reducing cross-presentation by a factor of four (Fig. 5b). This suggests that the SIINFEKL epitope is indeed liberated from the pool of dislocated ddRLuc-Fc$^{OVA}$ protein and subsequently bound to MHC-I molecules.

To further investigate the correlation between dislocation and cross-presentation, we treated 293T-FcγIIA-K$^b$ cells with chloroquine (CHQ), a drug previously shown to enhance both processes[40]. As expected, CHQ increased both ddRLuc-Fc$^{OVA}$ dislocation and cross-presentation in a dose-dependent manner (Supplementary Fig. 9a). Inhibition of VCP/p97 is also known to reduce dislocation (Figs. 2 and 4) and cross-presentation[11,19], and indeed CB5083 treatment of 293T-FcγIIA-K$^b$ cells inhibited ddRLuc-Fc$^{OVA}$-derived luminescence and T cell activation in a dose-dependent manner (Supplementary Fig. 9b). Moreover, inhibition of substrate internalization by Cyto D and Dyn caused an almost complete loss of luciferase activity and cross-presentation (Supplementary Fig. 9c). We combined all the data from these various drug treatments and produced a correlation curve, plotting luminescence in the presence of Epox against cross-presentation efficiency without it. Strikingly, we observed an excellent linear correlation between the two processes (Fig. 5c), suggesting that dislocation and cross-presentation are highly coupled and that dislocation is the rate limiting step in cross-presentation.

## Discussion

The experiments presented reinforce the concept that exogenous proteins can access the cytosol of DCs and certain other cell types. The ddRLuc-Fc substrate eliminates the drawbacks of approaches previously used to investigate this process[6,12,14,21,22,41–44]. Many require very high levels of the proteins, lack sensitivity, and produce a complex readout that is often hard to interpret. The requirement for deglycosylation by the cytosolic enzyme NGLY1 to generate luminescence, along with the dependence on proteasome inhibition, provides the essential evidence that ddRLuc-Fc enters the cytosol. Inhibition by reagents interfering with endocytosis and/or phagocytosis demonstrate that active internalization is essential, and the activities of Hsp90 and VCP/p97 are also required for the maximum signal. The addition of an Hsp90 inhibitor after peak luciferase activity is reached following phagocytosis results in accelerated decay of activity. This is consistent with Hsp90 facilitating cytosolic refolding and stabilization, while the lack of an effect when VCP/p97 is inhibited under identical circumstances is consistent with a role for this AAA ATPase in the initial entry of the enzyme into the cytosol, similar to its proposed role in ERAD. ddRLuc-Fc allows detection of cytosolic dislocation with high sensitivity in primary DCs of

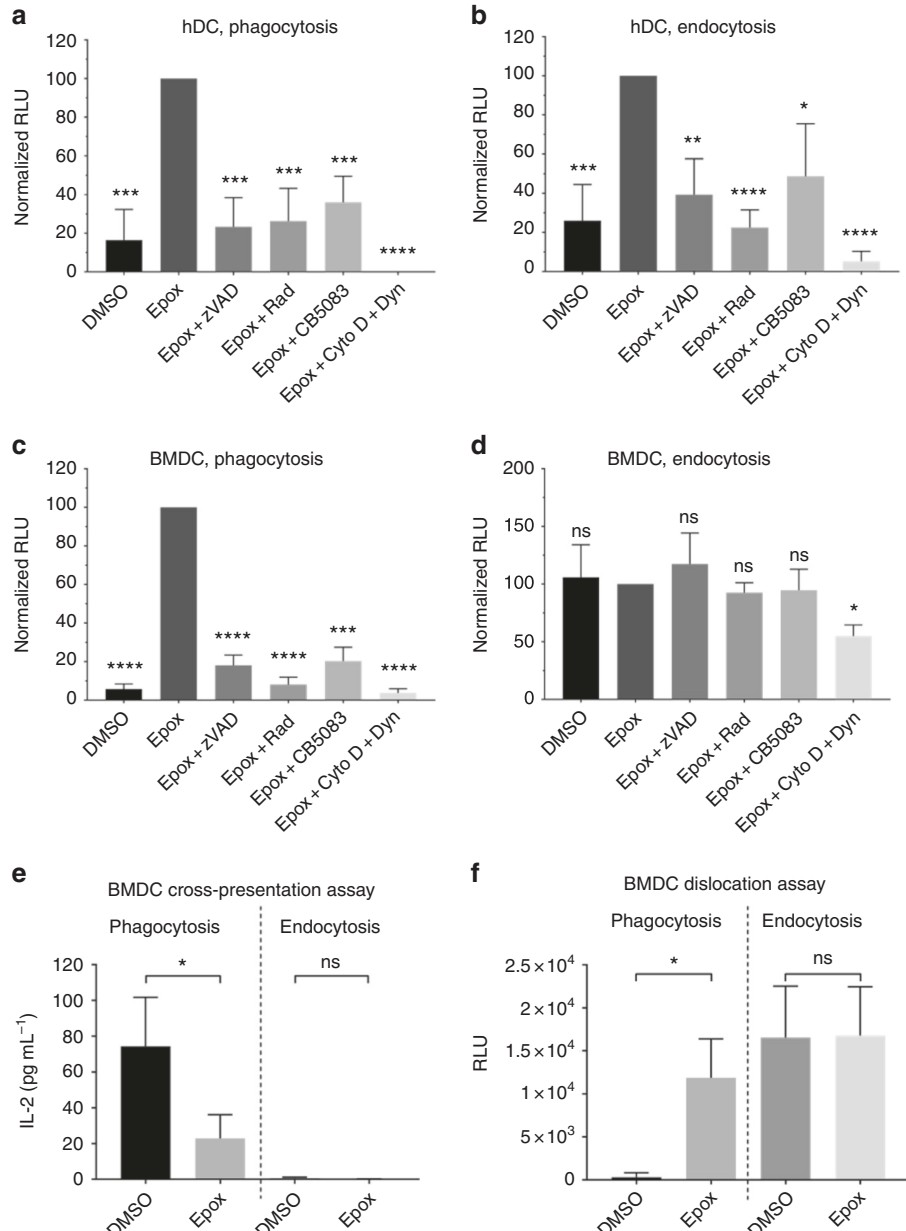

**Fig. 4** ddRLuc-Fc dislocation and cross-presentation by primary dendritic cells. **a–d** Primary dendritic cells were incubated with ddRLuc-Fc-bound 3 μm latex beads for phagocytosis (**a**, **c**) or soluble ddRLuc-Fc at 100 μg mL⁻¹ for endocytosis (**b**, **d**). Cell lysates were prepared, processed, and analyzed as described in Fig. 2. Human DCs (hDCs) (**a**, **b**) were harvested 6 h after phagocytosis/endocytosis, while mouse bone marrow-derived DCs (BMDCs) (**c**, **d**) were harvested after 3 h. Drugs were used at the following concentrations: 800 nM Epox, 20 μm zVAD, 32 μm Rad, 1 μm CB5083, 2.5 μg mL⁻¹ Cyto D plus 100 μm Dyn. **e**, **f** Mouse BMDCs were incubated with ddRLuc-Fc$^{OVA}$-bound 3 μm latex beads at a 30:1 bead:cell ratio for phagocytosis (left panels) or with 50 μg mL⁻¹ soluble ddRLuc-Fc for endocytosis (right panels) in the presence or absence of 800 nM Epox. Cells were harvested at 3 h, and assessed for luciferase activity (**e**) or stimulation of IL-2 secretion after an additional overnight incubation with the B3Z hybridoma (**f**). Bars represent the mean +/−s.d. of at least three independent experiments per treatment (paired two-tailed $t$-test, $*p < 0.05$; $**p < 0.01$; $***p < 0.001$; $****p < 0.0001$; ns, not significant)

mouse and human origin, in FcR expressing non-DCs, and even in FcR-negative cell lines.

ddRLuc itself is insufficiently stable for large-scale purification. Producing it as an Fc fusion protein facilitated affinity purification and in part solved this stability problem. However, the half-life of glycosylated ddRLuc-Fc at 37 °C (~20 min) is still extremely short compared to WT RLuc (~84 h)[27]. To a large extent, instability makes ddRLuc-Fc a good probe for the dislocation process. Glycosylated ddRLuc-Fc has a low level of enzymatic activity (Supplementary Table 2) and any residual extracellular activity decreases rapidly at 37 °C.

Deglycosylation inherently stabilizes it, increasing its half-life by fivefold, and in the cytosol this is further increased to 10–12 h. This prolonged activity allows the detection of the cytosolic enzyme for up to 24 h, which could facilitate identification of cells that have internalized the protein following in vivo injection.

In a new and striking finding we observed that, unlike in human DCs, ddRLuc-Fc dislocation in mouse BMDCs is undetectable after endocytosis, while both human and mouse DCs are highly efficient following phagocytosis. The lack of dislocation following endocytosis was paralleled by the lack of cross-presentation of ddRLuc-

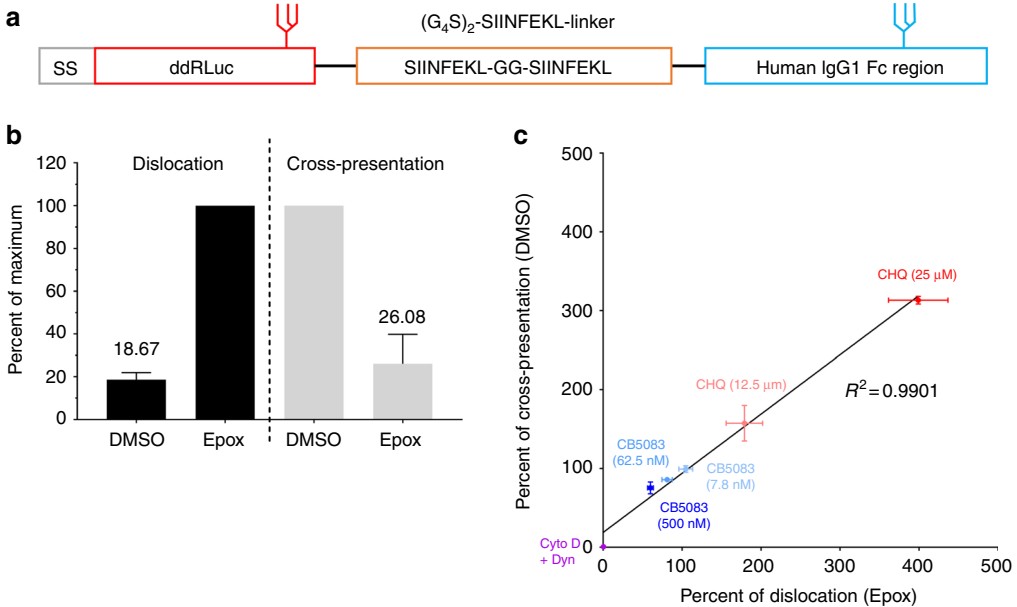

**Fig. 5** Correlation between antigen dislocation and T cell activation. **a** Schematic description of ddRLuc-Fc[OVA]. **b, c** The 293T-FcγIIA-K[b] cells were fed ddRLuc-Fc[OVA]-bound 3 μm latex beads for phagocytosis and incubated at 37 °C for 8 h. For measuring luciferase activity, total cell lysate was prepared, processed, and analyzed as described in Fig. 2. The percent RLU values shown are generated from samples treated with DMSO or a combination of Epox plus various inhibitors relative to Epox-treated samples (set to 100%). Cross-presentation was assayed using IL-2 secretion by the B3Z hybridoma as described in Methods. The percent cross-presentation shown corresponds to the amount of IL-2 released from the B3Z hybridoma activated by the cells treated with various inhibitors relative to DMSO-treated samples (set to 100%). **b** Opposing effects of proteasome inhibition by 200 nM Epox on dislocation and cross-presentation. Bars represent the mean +/−s.d. of five independent experiments per treatment, and numbers indicate the mean value. **c** The correlation between antigen dislocation and cross-presentation was plotted by incorporating data from treatments with different concentrations of chloroquine (CHQ), CB5083, and cytochalasin D (Cyto D) plus Dynasore (Dyn) in the presence (for dislocation) or absence (for cross-presentation) of Epox. Data from Supplementary Fig. 9 were also incorporated into this figure. Points represent the mean +/−s.d. (for both x and y values) of one representative experiment with triplicates. Representative data from three independent experiments are shown

Fc[OVA] by BMDCs (Fig. 4e, f, right). The dislocation machinery may not be delivered to nascent endosomes or it may be delivered but not activated. Signaling through pro-inflammatory receptors such as TLRs could guide the specialization of endosomes and manipulate the antigen dislocation process[37,45], an important area for future studies with this reagent. BMDCs predominantly express the inhibitory receptor FcγRIIB[46] and enhanced cross-presentation by FcγRIIB knockout BMDCs has been observed both in vitro and in vivo[47,48]. FcγRIIB-driven endocytosis favors the formation of nondegradative vesicular compartments in BMDCs that preserve rather than degrade internalized antigen for subsequent B cell priming[49]. However, we generated a non-FcR binding derivative of ddRLuc-Fc by eliminating the glycosylation site in the human Fc region[50,51], and this derivative (ddRLuc-Fc[CHO-]), while retaining function upon phagocytosis, still failed to work after endocytosis, suggesting that FcγRIIB signaling may not be involved in inhibiting dislocation in BMDCs (Supplementary Fig. 10).

Studies of dislocation in different cell lines show that many are also dislocation competent and therefore potentially capable of cross-presentation. In fact, previous studies showed that 293T-FcγRIIA-K[b] cells can do both[10]. Calculations using estimated ddRLuc-Fc specific activities, and based on Table 1 and Supplementary Table 2, suggest that the dislocation process is surprisingly efficient. Phagocytosis for 6 h in the presence of Epox allows hDCs to accumulate approximately $2 \times 10^7$ active RLuc molecules per cell. BMDCs accumulate $3 \times 10^7$ molecules per cell in 3 h. 293T-FcγRIIA-K[b]-cells accumulate $6 \times 10^8$ molecules per cell, and even FcR-negative 293T cells accumulate $1 \times 10^6$ molecules per cell in 6 h. Notably, these numbers refer only to active RLuc enzyme and the number of molecules that accumulate is likely to be higher when Epox-insensitive degradation and denaturation

are accounted for. Different cell types may prefer different internalization pathways. BMDCs and 293T-FcγRIIA-K[b] cells dislocate more efficiently following phagocytosis, while human DCs, the EBV-transformed cell line TRal, and FcR-negative Expi293F cells favor endocytosis (Table 1). The differences may reflect the efficiency of the different internalization pathways. For example, the low level of luminescence generation in 293T cells after both phagocytosis and endocytosis (Table 1) is mainly because of poor internalization. Alternatively, differences in protein degradation or stability within the downstream compartments and/or the recruitment of dislocation machinery may play a role. For example, endocytic internalization of ddRLuc-Fc by BMDCs is efficient, but it appears to accumulate within the cell without deglycosylation, perhaps because of superior stabilization in the cytosol or the endocytic pathway. Unraveling the biochemical and cell biological mechanisms underpinning these differences will be important going forward.

Using the same substrate (ddRLuc-Fc[OVA]) to quantify antigen dislocation and cross-presentation proved a powerful approach. By incorporating data obtained with a variety of pharmacological agents used at different concentrations, such as chloroquine and the VCP/p97 inhibitor CB5083, we were able to show a linear relationship between antigen dislocation into the cytosol and cross-presentation (Fig. 5c). Notably, 293T cells are not generally considered as immune cells and yet they clearly can mediate the dislocation process. This suggests that cytosolic dislocation of exogenous proteins could be important for processes other than cross-presentation.

The versatility and sensitivity of the reagents we have developed opens many possibilities for future applications. These include screens using siRNA or CRISPR/cas9 to identify factors

involved in protein dislocation, the discovery and characterization of novel cross-presenting cell subsets in vivo, and the identification of phagocytic receptors other than Fc receptors that regulate the dislocation process. Moreover, the tool may have clinical significance in that it could potentially be used to monitor drug delivery into the cytosol, to assess the potential efficiency of adjuvants to initiate cross-presentation, or for drug screening approaches to identify stimulators or inhibitors of cross-presentation that may have therapeutic value.

## Methods

**Antibodies and reagents.** Mouse anti-RLuc mAb was from EMD Millipore Corporation (Cat# MAB4410) and used at 1:5000 dilution for western blotting (WB). Rat anti-GRP94 mAb was from Enzo Life Sciences (Cat# ADI-SPA-850-D) and used at 1:5000 dilution for WB. Rabbit anti-prosaposin mAb was from Abcam (Cat# ab166910) and used at 1:50,000 dilution for WB. Mouse anti-GAPDH mAb was from Invitrogen/Thermo Fisher (Cat# AM4300) and used at 1:5000 dilution for WB. Goat anti-Cathepsin D pAb was from Santa Cruz Biotechnology (Cat# sc-6486) and used at 1:200 dilution for WB. Renilla luciferase and related reagents were purchased from Prolume, AZ. Recombinant Renilla luciferase (Cat# 312) was dissolved in PBS, native Coelenterazine (nCTZ) (Cat# 303) and e-Coelenterazine (e-CTZ) (Cat# 355) were dissolved in ethanol at 1 mg mL$^{-1}$ and diluted 1:500 in PBS for luciferase assays or luminescence microscopy. Streptolysin O (SLO) was from Sigma-Aldrich (Cat# S5265). Epoxomicin was from Enzo Life Sciences (Cat# BML-PI127-0100) or ApexBio Technology (Cat# A2606). z-VAD(OMe)-FMK was from Abcam (Cat# ab120487). Cytochalasin D was from Sigma-Aldrich (Cat# C8273). Dynasore was from Abcam (Cat# ab120192). Radicicol was from AG Scientific (Cat# R-1130). CB5083 was from Selleck Chemicals (Cat# S8101). Chloroquine diphosphate was from Sigma-Aldrich (Cat# C6628). Passive Lysis Buffer, 5× (Cat# E194A) and luciferin (Cat# E1501) were from Promega.

**Cells and cultures.** The 293T cells were originally purchased from ATCC (American Type Culture Collection, USA). The 293T-FcγRIIA-K$^{b}$[10] and the TRal[36] cells were made in our laboratory. The 293T, 293T-FcγRIIA-K$^{b}$, and TRal cells were cultured at 37 °C and 5% CO$_2$ in Iscove's Modified Dulbecco's Medium (IMDM) (Sigma-Aldrich, Cat# I3390), supplemented with 10% FBS (Gemini Bio-Products, Cat# 100-106), 1x GlutaMAX (Thermo Fisher, Cat# 35050061), and 1x Penicillin-Streptomycin (Pen-Strep) (Thermo Fisher, Cat# 15140122).

The Expi293F™ cells (Thermo Fisher, Cat# A14527) were cultured at 37 °C and 8% CO$_2$ in Expi293™ Expression Medium (Thermo Fisher, Cat# A1435102), and shaken at 130–150 rpm on Microplate Shaker (VWR, Cat# 12620).

To generate BMDCs, mouse bone marrow cells were isolated from pathogen-free C57BL/6 mice (6–10 weeks old), and cultured at 37 °C and 5% CO$_2$ in RPMI-1640 (Thermo Fisher, Cat# 11875119), supplemented with 10% FBS, 2 mM L-Glutamine (Thermo Fisher, Cat# 25030081), 1x Pen-Strep, 1x non-essential amino acids (Thermo Fisher, Cat# 11140076), 15 mM HEPES (Thermo Fisher, Cat# 15630080), 50 μm 2-mercaptoethanol (β-ME) (Thermo Fisher, Cat# 21985023), and 1% conditioned media derived from the mouse granulocyte-macrophage colony stimulating factor (GM-CSF) producing cell line J577L (obtained from Ruslan M. Medzhitov, Yale University, USA). Immature BMDCs were harvested on day 6.

To generate human DCs, CD14$^{+}$ mononuclear cells were isolated from buffy coat blood using RosetteSep™ Human Monocyte Enrichment Cocktail (STEMCELL Technologies, Cat# 15028) according to the manufacturer's protocol. Cells were cultured at 37 °C and 5% CO$_2$ in RPMI-1640, supplemented with 10% FBS, 2 mM L-Glutamine, 1x Pen-Strep, 25 ng mL$^{-1}$ recombinant human GM-CSF (Leukine (sargramostim), Sanofi) and 10 ng mL$^{-1}$ recombinant human IL-4 (PeproTech, Cat# 200-04). The DCs were harvested on day 6.

The H2-K$^{b}$-OVA-specific B3Z hybridoma[52] was cultured at 37 °C and 5% CO$_2$ in RPMI-1640, supplemented with 10% FBS, 25 mM HEPES, 1 mM sodium pyruvate (Thermo Fisher, Cat# 11360070), 1x Pen/Strep, and 50 μm β-ME.

**Expi293F cell transfection and protein purification.** Approximately, 1 mg plasmid encoding ddRLuc-Fc or its derivatives was transfected into 1 L Expi293F cells at 1 × 10$^6$ per mL using linear Polyethylenimine (PEI) (Polysciences, Cat# 23966) at 3:1 (PEI:DNA) mass ratio. Furthermore, 24 h post transfection, protein expression enhancer cocktail consisting of 1 mM valproic acid (Sigma-Aldrich, Cat# P4543), 6 mM sodium propionate (Sigma-Aldrich, Cat# P1880), and 0.5% tryptone (AmericanBio, Cat#: AB02031) were added to cells to increase protein yield. Cells were harvested 60 h after transfection.

Pelleted cells were lysed in TBS (10 mM Tris, 150 mM NaCl, pH 7.6) containing 0.5% Triton X-100 (AmericanBio, Cat# AB02025) and EDTA-free Protease Inhibitor Cocktail (Roche, Cat# 11873580001) at 4 °C for 45 min, followed by centrifugation at 110,000 × g in an Optima L-100K ultracentrifuge (Beckman Coulter) for 1 h at 4 °C. The lysate was filtered through a 0.22 μm filter (EMD Millipore, Cat# SCGPT01RE) to remove cell membranes and large aggregates. Furthermore, 1 mL of protein A-Sepharose beads (GE Healthcare, Cat# 17-0780-01) were added to the lysate and after rotation at 4 °C for 2–3 h. The beads were

washed three times in TBS before protein elution in 20 mL 100 mM glycine-HCl buffer, pH 3.3. The eluate was neutralized by adding 1 M Tris pH 9. Purified protein was concentrated, and buffer was exchanged to PBS using Zeba™ Spin Desalting Column (Thermo Fisher, Cat# 87769). Protein purity was determined by Coomassie blue staining and concentration was determined by quantitative WB in comparison to commercially available recombinant RLuc (Prolume, Cat# 312).

**Luciferase assay.** Samples were added to a 96-well white plate (Berthold, Cat# 23300). A Centro XS³ LB 960 High Sensitivity Microplate Luminometer (Berthold, Cat# 46970) was used for luminescence detection. 100 μL diluted nCTZ or 50 μL luciferin solution was injected into each well for RLuc and FLuc luminescence generation, respectively. Luminescence was measured for 1 s.

**Deglycosylation in vitro and ddRLuc-Fc stability.** For deglycosylation, purified ddRLuc-Fc was incubated at 37 °C for 10 min in the presence of Endo H (NEB, Cat# P0702) or PNGase F (NEB, Cat# P0704) according to the manufacturer's protocol. To determine the stability of glycosylated ddRLuc-Fc, it was incubated at 37 °C for different length of time followed by PNGase F digestion at 37 °C for 10 min before luciferase assay. To determine the stability of deglycosylated ddRLuc-Fc, the purified protein was continuously incubated with PNGase F at 37 °C, with an additional 10 min incubation at the end of each time point to match the 10 min deglycosylation required to determine the stability of the glycosylated enzyme.

**Luminescence microscopy.** An LV200 Inverted Microscope (OLYMPUS) was used for luminescence microscopy. In Fig. 1c, transfected 293T cells in a 24-well plate were first washed three times with 0.5 mL PBS, 0.5 mL nCLZ was added and images were taken with a 200 s exposure. The luminescence generated from dislocated ddRLuc-Fc in 293T-FcγRIIA-K$^{b}$ cells after phagocytosis required an enhanced RLuc substrate, e-CLZ, and the images required a 5 min exposure (Fig. 2c). Brightness bars reflecting relative values ranging from 0 to 255 on a linear scale were added to the lower left side of each luminescence image, calculated using the ImageJ program (https://imagej.nih.gov).

**The dislocation assay.** Cells were plated in 96-well V-bottom plate and incubated with different combinations of inhibitors at the specified concentration for 30 min at 37 °C before adding the substrate. DMSO was used as the solvent of all inhibitors and used at less than 0.5% as the control treatment in all experiments. For phagocytosis, ddRLuc-Fc was non-covalently bound to 3 μm latex beads (Polysciences, Cat# 17134) by rotating at 4 °C for 12 h. The washed beads were added to the cells at a 50:1 (bead:cell) ratio unless specified. For endocytosis or fluid-phase macro-pinocytosis, soluble ddRLuc-Fc was added to cells at a final concentration of 100 μg mL$^{-1}$ unless specified. For kinetic experiments the cells were incubated with beads or soluble enzyme for 30 min on ice before washing and warming to 37 °C to initiate internalization. After incubation the cells were washed three times with ice cold PBS and lysed in 50 μL 1x passive lysis buffer supplemented with 20 μM zVAD (to avoid post-lysis deglycosylation) at room temperature (RT) for 20 min. Furthermore, 10 μL total cell lysate was then directly assayed for luciferase activity. PNGase F digested cell lysate (8 μL) was also analyzed to determine the total internalized ddRLuc-Fc activity/amount.

**Streptolysin O treatment.** After incubation with 3 μm latex beads coated with ddRLuc-Fc for 8 h, 293T-FcγRIIA-K$^{b}$ cells (5 × 10$^5$) were washed three times with IMDM and permeabilized with SLO as previously described[35]. Briefly, cells were resuspended in 120 μL IMDM and incubated on ice for 15 min in the presence of 0.3 mM DTT with or without SLO at 20 μg mL$^{-1}$. Cells were washed once with 0.5 mL DPBS and then resuspended in 120 uL DPBS followed by incubation at 37 °C for 5 min to allow permeabilization. Cells were pelleted at 300 × g for 5 min at 4 °C and the supernatant was transferred to ultra-centrifuge tube followed by centrifugation at 100,000 × g for 1 h at 4 °C to remove any cellular membranes. The cytosol was examined by western blotting for Grp94, prosaposin, cathepsin D, and GAPDH and assayed for luciferase activity as described above.

**Cross-presentation assay.** The 293T-FcγRIIA-K$^{b}$ cells (1 × 10$^5$ per well) or BMDCs (2 × 10$^5$ per well) were distributed in 96-well flat bottom plates. ddRLuc-Fc$^{OVA}$-bound 3 μm latex beads or 50 μg mL$^{-1}$ soluble ddRLuc-Fc$^{OVA}$ was added in the presence of various inhibitors. After incubation at 37 °C, cells were fixed in 0.5% paraformaldehyde for 293T-FcγRIIA-K$^{b}$ cells or 1% paraformaldehyde for BMDCs at 37 °C for 10 min. Fixation was stopped with 200 mM glycine dissolved in PBS, pH 7.5, followed by two washes with BMDC medium, and one wash with B3Z medium. A total of 5 × 10$^4$ B3Z T cell hybridoma cells were added per well and after 15 h incubation, released IL-2 was determined by ELISA, according to the manufacturer's protocol (BD Biosciences, Cat# 555148).

**Statistical analysis.** Individual experiments were performed in 2–4 replicates. For luminescence generated in RLuc-transfected 293T cells, relative luciferase units (RLU) were normalized to the RLU generated by co-transfected FLuc. For luminescence generated in the dislocation assay, RLU was normalized to total protein in the cell lysate measured by the Bradford protein assay. For studies involving drug

treatment, the RLU from cell-only control (background) was subtracted from the RLU of each treatment and normalized to the value of Epox-treated samples, which were set to 100. In each independent experiment, average values of at least two replicates of different treatments were compared. Two-tailed $p$ value was calculated by paired student $t$-test with the assumption of Gaussian distribution (GraphPad Prism 7) and used to determinate the statistical significance: $*p < 0.05$; $**p < 0.01$; $***p < 0.001$; $****p < 0.0001$; ns, not significant.

**Data availability**. The data that support the findings of this study are available from the corresponding author upon reasonable request.

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

## Acknowledgements

We thank Dr. Jun Wang for kindly providing the human IgG1 Fc vector, Dr. David Schatz for providing Expi293F cells and Dr. Aaron Ring for the protein expression enhancer cocktail recipe. We are especially grateful to Dr. Kiran Nataraj for his expertize and to Olympus for the loan of the LV200 Inverted luminescence microscope. The work was supported by NIH grants RO1-AI097206 and R01-AI059167 awarded to P.C.

## Author contributions

Q.L., J.E.G., and P.C. designed the project, analyzed the results, and wrote the manuscript. Q.L. performed all experiments.

## Additional information

**Competing interests:** The authors declare no competing interests.

