## [Peer Review File · Nature Communications]

Reviewers' comments:

Reviewer #1 (Remarks to the Author):

In their manuscript, the authors describe the development of a novel tool to monitor the dislocation of extracellular antigens into the cytoplasm. Such dislocation is a crucial step in antigen cross-presentation. The molecular mechanisms enabling antigen dislocation into the cytosol are not fully understood. Therefore, the field would benefit from the development of such a tool.

In their study, the authors developed a glycosylated luciferase mutant, which becomes active after dislocation of deglycosylation in the cytosol. The study is well designed and the idea of generating a glycosylation-dependent luciferase innovative. The experiments shown are convincing and statistics are fine.

However, to my opinion, some of the conclusions are not justified.

Major concerns:

- If the activation of the luciferase indeed is due to deglycosylation after antigen dislocation (which is the main point of the paper), this needs to be shown directly. Therefore, luciferase from cytosolic fractions (using SLO as already performed in the study) and from the remaining fraction (Endosomes,...) should be analyzed by SDS Page / Western Blot to demonstrate actual differences in glycosylation. If luciferase concentrations should be too low to visualize, it should be possible to concentrate them using Protein A/G based affinity chromatography (using the Fc part of the protein)

- Table 1 demonstrates that a clear luciferase signal can only be detected after proteasome inhibition. However, using soluble luciferase on murine BM-DCs, the signal is not increased by epoxomicin. The authors suggest that this might be due to increased uptake, increased intra-endosomal stability of the antigen or greater cytosolic accumulation without deglycosylation. However, if this would be true, the luciferase shouldn't be active. But absolute numbers in table 1 demonstrate much higher luciferase activity in BM-DCs after treatment with soluble luciferase. If this would be still endosomal luciferase, it should be either 1) glycosylated and therefore inactive or 2) deglycosylated in the endosomes. If 1) would be true, the very high signal would be background signal from huge amounts of endosomal luciferase. In this case, it would be impossible to visualize small amounts of dislocated luciferase in the cytoplasm, which would make the luciferase not suited for experiments in these cells. If 2) would be true and the luciferase can be deglycosylated in endosomes of these cells, the luciferase would also not be suited to be used in these cells.

If the high signal in table 1 would be due to increased amounts of still glycosylated luciferase in the cytosol, the luciferase should not be active (because glycosylated) and the observed amounts of luciferase in table 1 would be background staining, which would also make the luciferase non-suited for experiments in these cells.

Taken together, the conclusion from these experiments should be that, using this tool, no conclusions on the molecular mechanisms of cross-presentation of soluble antigens can be made in these cells at all.

- Additionally, the authors than point out that in cross-presentation experiments from other groups, very high concentrations of OVA were used. They suggest that, using lower concentrations, cytosolic dislocation might not be involved in endocytic cross-presentation. Additionally, they showed an experiment using lower concentrations of antigen (ddRLuc-FcOVA) and could not observe cross-presentation or dislocation. They suggest from these results that in experiments from other groups, high concentrations of OVA might have lead to aggregates and rather phagocytosis, and insinuate that cross-presentation of soluble antigens doesn't occur. I agree that there are publications using high antigen concentrations, but still, these are clear overinterpretations that are in contradiction with a significant amount of literature, including publications from the Cresswell group as well. In Singh and Cresswell (Science 2010), they demonstrated clear cross-presentation of soluble OVA by murine BM-DCs even at concentrations

as low as 0,001 mg/ml. Therefore, they cannot insinuate that cross-presentation of soluble antigens in lower concentrations doesn't occur! Of course, the vacuolar pathway of cross-presentation allows dislocation-independent cross-presentation but still, a substantial amount of literature proves that cross-presentation of soluble antigens is often TAP and proteasome dependent. This should be clarified in the text. Additionally, as stated above, conclusions on the molecular mechanisms of cross-presentation in murine BM-DCs cannot be drawn at all from the experiments shown here.

- As the authors state correctly in their manuscript, pro-inflammatory receptors like TLRs might influence dislocation. However, they generated their luciferase as a fusion construct to the Fc receptor. Since it is known that activation of Fc receptors activates DCs, they might already look at an activated cell type. In fact, cross-linking of Fc receptors will have different effects on DC activation than binding of monovalent ligands. The authors should exclude that differences in phagocytosed / endocytosed luciferase might be due to different DC activation after Fc receptor cross-linking. Especially since the authors already stated that Fc-luciferase tended to activate BM-DCs.

Reviewer #2 (Remarks to the Author):

The paper by Lu et al describe a luciferase probe designed to detect cytosolic penetration, a step of importance in antigen cross presentation. The data presented revolves around the characterization of the probe designed and provide some insights on the translocation process.

Here are my comments (not by order of importance):

1) This article is generally hard to follow, in part because of the back and forth between Figures and supplemental material. However, a number of things make it also more confusing than necessary. For instance, it is unclear why the y-axis on 1b is "%Epox" as opposed to "normalized luminescence". Figure 5 is presented before figure 4 e and f. Supplemental data Figure 9 is presented in Discussion as opposed to results. Several conclusions are reached without being clearly outlined step by step (see below).

I'm not sure that the term "dislocation" is right here...translocation, as used in the intro, seems more appropriate.

2) Figure 1. Figure 1 c should show bright field images to illustrate that cells are present, even when no luminescence is observed. The luminescence could also be pseudocolored by intensity to better show the differences in signal. Given the lengthy treatment with inhibitors, the viability of the cells should also be reported.

3) Figure 2/3. The authors use DMSO as a control, and I presume that the assumption is that DMSO has no effect (it's not clear how much is used and it is not clear how much is present in Epox or sVAD stocks). Untreated cells should be included for comparison (untreated cells and DMSO-treated cells could be different). While it is likely that Epox has an inhibitory effect on cytosolic proteasome, absence of an effect on endosomal degradation should ideally be demonstrated. The authors should also provide additional evidence that the activation of the probe, or its release from endosomes, does not occur during cell lysis (the authors add zVAD in the lysis buffer but do not provide evidence that this is effective under these conditions). The cells treated with the combinations of inhibitors should be tested for viability and proliferation. This is because apoptosis can mediate the permeabilization of membranes. In particular, a concern with this assay may be that a few cells (dying cells) could cause the release and activation of a lot of probe. Figure 2c addresses this concern in part. However, it is only qualitative and does not rule out the possibility that a few super luminescent cells contribute to the majority of the signal detected in the lysate.

What about the possibility that NGLY1 is released by dead cells and that it activates the probe during the incubation time (prior or during endocytosis)?

The kinetics in Figure 3b are significantly different than 3a, in particular in how it responds to inhibitors. why is the baseline at 20% even when there is no protein? Cell viability/stress after long term exposure to inhibitors?

3) Supp info 3a.: the authors test the purity of the cytosolic fraction by showing exclusion of lysosomal proteins. This approach should be extended to other endocytic organelles: early endosomes, late endosomes, multivesicular bodies, recycling endosomes. This is necessary to show that the probe release after the treatment with SLO does not come from endosomally trapped material.

4) "this strongly supports a role for VCP/p97...": this conclusion is unclear and requires a comparison between Fig2 and Supp Fig 4...see comment 1.

5) Figure 4e. Is it possible that phagosomes/endosomes containing beads are more easily disrupted during lysis than regular endosomes?

6) several probes have been reported in the CPP field to detect cytosolic penetration. The authors should mention some of this work in their intro and in their discussion.

Reviewer #3 (Remarks to the Author):

This manuscript reports the development and use of a recombinant protein that can be used to detect and quantify the translocation of proteins from vesicular compartments into the cytosol of cells. The approach is really ingenious and elegant. Assays using this probe have an excellent signal-to-noise ratio. The authors do an excellent job of establishing the validity of their assay. Current reagent/approaches to measure vesicle-to-cytosol translocation are not very good, so the author's system is a major technical advance that will be useful to the field. The authors use their probe to quantify and characterize the translocation in various cells and in cross presentation with interesting results. They show that more cells can translocate internalized proteins, albeit with very different efficiencies, than were generally thought and document differences in this process in different dendritic cells. They give insight into the kinetics of this translocation. They confirm the role of HSP90 and p97 in this process giving insight into mechanism. Another important finding is a remarkable correlation between dislocation and antigen presentation suggesting the antigen transfer is a rate limiting in cross presentation.

There are some minor points that the authors may want to consider but are not essential to address.

1. The authors quantify the number of molecules of probe that are translocated into the cytosol. It would be of interest to quantify the amount of probe that was internalized and calculate the percent translocated.

2. In Fig 3, many of the groups were incubated with proteasome inhibitors for long periods (up to 24 hours) and with many cell types (I don't know about 293t) this will lead to a loss of viability. While this is probably not a problem at the early time points (which are the most informative ones), it would be useful to know whether viability was affected over the time course of the experiment? Any effect of derangement of cell function might be assessed by pre-treating with proteasome inhibitors for various times before adding the probe and seeing if the dislocation kinetics are altered (provided the proteasome inhibitor doesn't reduce the internalization of the probe).

3. It would be of interest to compare the kinetics of dislocation (Fig 3) with the kinetics of cross-presentation.

4. A casual reader may look at graphs, e.g. Fig 2a and 2b, and interpret them to show an inhibition by DMSO. The authors might want to consider moving this group to be the first bar and labeling it "none" or "control" on the graph, and also somehow showing that the statistically significant change is an increase in the epox group (not a decrease from DMSO); e.g. separate horizontal bars over the DMSO vs Epox and Epox-vs other groups. The authors might want to include data somewhere comparing DMSO to just media, which I presume are not different.

Reviewer

Kenneth L. ROCK

Response to reviewers.

We thank the reviewers for their thoughtful comments. We have introduced new data and experiments to address their concerns, and we discuss the issues raised in the text and in the comments below. Changes to the text of the manuscript in response to the reviewers' comments are highlighted in yellow.

As a preamble, it seems that some of the reviewers are concerned that the luciferase probes might somehow be converted to an active form in the endocytic pathway. I would point out that the only mammalian enzyme that can mediate deglycosylation resulting in the asparagine to aspartyl conversion required for luciferase activity is NGLY1 and that this is a cytosolic enzyme. Endocytic deglycosylation occurs by successive removal of sugars and the amino acid conversion does not occur. We have introduced new data in response to the reviewers' concerns, plus one new experiment that addresses the issue of whether the deglycosylated enzyme is in the cytosol, as well as addressing the mechanism by which its activity is maintained. We show that the addition of radicicol to 293T-FcRγIIa-K^b cells that have phagocytosed ddRLuc-Fc after peak deglycosylation-dependent luciferase activity is reached induces its accelerated decay, consistent with our hypothesis that the enzyme is stabilized by cytosolic Hsp90. This is presented in Supplementary Figure 7.

Specific responses to the individual reviewers' comments follow:

Reviewer #1 (Remarks to the Author):

In their manuscript, the authors describe the development of a novel tool to monitor the dislocation of extracellular antigens into the cytoplasm. Such dislocation is a crucial step in antigen cross-presentation. The molecular mechanisms enabling antigen dislocation into the cytosol are not fully understood. Therefore, the field would benefit from the development of such a tool.

In their study, the authors developed a glycosylated luciferase mutant, which becomes active after dislocation of deglycosylation in the cytosol. The study is well designed and the idea of generating a glycosylation-dependent luciferase innovative. The experiments shown are convincing and statistics are fine.

However, to my opinion, some of the conclusions are not justified.

Major concerns:

1) If the activation of the luciferase indeed is due to deglycosylation after antigen dislocation (which is the main point of the paper), this needs to be shown directly. Therefore, luciferase from cytosolic fractions (using SLO as already performed in the study) and from the remaining fraction (Endosomes,...) should be analyzed by SDS Page / Western Blot to demonstrate actual differences in glycosylation. If luciferase concentrations should be too low to visualize, it should be possible to concentrate them using Protein A/G based affinity chromatography (using the Fc part of the protein)

This is an excellent suggestion and in fact we had already tried it. However, our experiments using protein A-Sepharose pull downs followed by western blots were not successful, even when we used large numbers of cells. Clearly luciferase activity is much more sensitive than a western blot. As an alternative approach, we used SLO permeabilization to isolate cytosol from the 293T-FcRγIIa-K^b cells that had phagocytosed ddRLuc-Fc in the presence of epoxomicin and then determined the proportion that

was glycosylated based on binding or lack of binding to Concanavilin A-Sepharose, a lectin specific for N-linked glycan. The results, shown in Supplementary Fig. 3d and e, and described in the text at the end of the first paragraph on p. 4, show that in the absence of zVAD approximately 70% of the cytosolic enzyme is deglycosylated, while when zVAD present this is reduced to less than 10%. This data supports the concept that deglycosylation occurs in the cytosol. Furthermore, if NGLY1 existed in the endocytic pathway, we should be able to detect deglycosylation-dependent luciferase activity in BMDCs following endocytosis (Fig 4d), but we do not.

2) Table 1 demonstrates that a clear luciferase signal can only be detected after proteasome inhibition. However, using soluble luciferase on murine BM-DCs, the signal is not increased by epoxomicin. The authors suggest that this might be due to increased uptake, increased intra-endosomal stability of the antigen or greater cytosolic accumulation without deglycosylation. However, if this would be true, the luciferase shouldn't be active. But absolute numbers in table 1 demonstrate much higher luciferase activity in BM-DCs after treatment with soluble luciferase. If this would be still endosomal luciferase, it should be either 1) glycosylated and therefore inactive or 2) deglycosylated in the endosomes. If 1 would be true, the very high signal would be background signal from huge amounts of endosomal luciferase. In this case, it would be impossible to visualize small amounts of dislocated luciferase in the cytoplasm, which would make the luciferase not suited for experiments in these cells. If 2) would be true and the luciferase can be deglycosylated in endosomes of these cells, the luciferase would also not be suited to be used in these cells.

If the high signal in table 1 would be due to increased amounts of still glycosylated luciferase in the cytosol, the luciferase should not be active (because glycosylated) and the observed amounts of luciferase in table 1 would be background staining, which would also make the luciferase non-suited for experiments in these cells.

Taken together, the conclusion from these experiments should be that, using this tool, no conclusions on the molecular mechanisms of cross-presentation of soluble antigens can be made in these cells at all.

I understand the reviewer's concerns but I would argue that the results of the experiment do not make the tool unsuitable for analysis of BMDCs. While the signal to noise ratio of ddRLuc-Fc is excellent, it is not perfect in that there is some residual activity when it is glycosylated. In fact, even in 293T-FcγIIa-K^b cells there is a signal that is independent of glycosylation immediately following endocytosis (see the 1 hour time point in Fig. 3b). BMDCs preserve this signal more effectively than other cell types after endocytosis but not phagocytosis. This is an extremely interesting mechanistic detail that we plan to explore, but we do not believe it is relevant to the current manuscript that seeks to establish the value of ddRLuc-Fc as a tool. The reviewer's sentence underlined above indicates what we believe to be the case; that the signal likely derives from substantial amounts of glycosylated enzyme present in the endocytic pathway of the cells that have not been dislocated into the cytosol for deglycosylation by NGLY1. Phagocytosis gives approximately 10⁴ RLU in the presence of epoxomicin but virtually zero in its absence (Fig. 4e). Endocytosis gives equal activity whether or not epoxomicin is added, arguing that what is seen is not dependent on proteasome inhibition and therefore corresponds to enzyme not susceptible to proteasome degradation, and therefore probably, but not definitively, still in the endocytic pathway. Additional activity equivalent to that resulting from phagocytosis that

depends on proteasome inhibition would be readily observable above the 2×10^4 RLU seen after endocytosis. Notably, however, Fig. 4f shows that, without epoxomicin addition, cross-presentation is also not detectable when the epitope-containing version of ddRLuc-Fc is used. Thus the reciprocal correlation between luciferase activity dependent on proteasome inhibition and cross-presentation in the absence of proteasome inhibition holds true. We believe the value of this manuscript lies in showing that the tool is useful as a probe for cytosolic access. We do not propose that the experiments presented explain the myriad molecular details of cross-presentation, but we do anticipate that it will facilitate addressing these issues.

4) Additionally, the authors than point out that in cross-presentation experiments from other groups, very high concentrations of OVA were used. They suggest that, using lower concentrations, cytosolic dislocation might not be involved in endocytic cross-presentation. Additionally, they showed an experiment using lower concentrations of antigen (ddRLuc-FcOVA) and could not observe cross-presentation or dislocation. They suggest from these results that in experiments from other groups, high concentrations of OVA might have lead to aggregates and rather phagocytosis, and insinuate that cross-presentation of soluble antigens doesn't occur. I agree that there are publications using high antigen concentrations, but still, these are clear overinterpretations that are in contradiction with a significant amount of literature, including publications from the Cresswell group as well. In Singh and Cresswell (Science 2010), they demonstrated clear cross-presentation of soluble OVA by murine BM-DCs even at concentrations as low as 0,001 mg/ml. Therefore, they cannot insinuate that cross-presentation of soluble antigens in lower concentrations doesn't occur! Of course, the vacuolar pathway of cross-presentation allows dislocation-independent cross-presentation but still, a substantial amount of literature proves that cross-presentation of soluble antigens is often TAP and proteasome dependent. This should be clarified in the text. Additionally, as stated above, conclusions on the molecular mechanisms of cross-presentation in murine BM-DCs cannot be drawn at all from the experiments shown here.

We congratulate the reviewer on using our own experiments against us and for pointing out the inconsistencies between these and others' experiments and those presented here! We agree that this part of the manuscript was overstated and we have revised it, beginning on line 8 of p. 6 and continuing through the second paragraph. However, we would point out that the experiments described rely on a short 3-hour exposure of the BMDCs to ddRLuc-Fc^{OVA} before the cells are fixed and exposed to the B3Z hybridoma. We chose this approach to avoid any potential activation of the BMDCs, which can affect cross-presentation. Our *Science* experiments cited above using low concentrations of OVA did not use fixation and involved overnight incubation in the presence of the hybridoma after a prior exposure of the BMDCs to OVA for 6 hours. In support of our data, early joint experiments by the Amigorena and van Endert groups that also used short time periods of incubation with soluble OVA and the same hybridoma obtained similar results, i.e. no cross-presentation, and wrote that '*Incubation of DC in the presence of soluble OVA without electroporation resulted in low levels of cross-presentation in intermediate DC and no cross-presentation in immature and mature DC*'. We cite this paper as ref 37. To ensure that our current results do not simply reflect the use of ddRLuc-Fc^{OVA}, we did similar experiments using BMDCs and soluble OVA and showed no cross-presentation after a 3 hour incubation, exactly as is the case with ddRLuc-Fc^{OVA}, while phagocytosis for the same period of time was very effective. Prolonging the incubation with soluble OVA did result in detectable epoxomicin-sensitive cross-presentation, which could be a result of BMDC maturation. We did not include these data in the

manuscript because they recapitulate the early data from van Endert and Amigorena, but they are presented below for the reviewer.

In addition, we showed that the difference in cytosolic entry after endocytosis or phagocytosis was recapitulated using ddRLuc-Fc lacking the Fc glycan, which does not bind to Fc receptors (Supplementary Fig. 10). This obviously does not end the debate, but does begin to address it.

5. As the authors state correctly in their manuscript, pro-inflammatory receptors like TLRs might influence dislocation. However, they generated their luciferase as a fusion construct to the Fc receptor. Since it is known that activation of Fc receptors activates DCs, they might already look at an activated cell type. In fact, cross-linking of Fc receptors will have different effects on DC activation than binding of monovalent ligands. The authors should exclude that differences in phagocytosed / endocytosed luciferase might be due to different DC activation after Fc receptor cross-linking. Especially since the authors already stated that Fc-luciferase tended to activate BM-DCs.

We completely agree with the reviewer that this is a subject worth addressing. However, it is a mechanistic question that lies beyond the scope of the present manuscript, which is concerned with establishing the ddRLuc system as a tool to study the translocation process. Notably, as pointed out above, the lack of cytosolic entry of soluble ddRLuc-Fc^{OVA} by BMDCs, measured by luciferase activity, correlates perfectly with lack of cross-presentation, while human DCs and the 293T derivative have no such problem. Also, as pointed out above, Supplementary Fig. 10 shows the same pattern of results for ddRLuc-Fc lacking the Fc glycan. I think the reviewer would agree that the mechanistic underpinnings of these differences need to be understood and that our ddRLuc system will be an excellent tool to figure this out.

Reviewer #2 (Remarks to the Author):

The paper by Lu et al describe a luciferase probe designed to detect cytosolic penetration, a step of importance in antigen cross presentation. The data presented revolves around the characterization of the probe designed and provide some insights on the translocation process.

Here are my comments (not by order of importance):

1) This article is generally hard to follow, in part because of the back and forth between Figures and supplemental material. However, a number of things make it also more confusing than necessary. For instance, it is unclear why the y-axis on 1b is "%Epox" as opposed to "normalized luminescence". Figure 5 is presented before figure 4 e and f. Supplemental data Figure 9 is presented in Discussion as opposed to results. Several conclusions are reached without being clearly outlined step by step (see below).

I'm not sure that the term "dislocation" is right here...translocation, as used in the intro, seems more appropriate.

We apologize for the lack of clarity, the complexity of the manuscript, and the requirement for numerous supplemental figures. Unfortunately, because of our attempts to ensure that all potential technical issues are addressed the supplemental figures are unavoidable. We have followed the reviewer's suggestions regarding the figures to improve their clarity, changing the order of the bars in the histograms to put the DMSO control first and the epoxomicin second, and by relabeling the axes essentially as suggested.

Supplementary Figure 10 (previously 9) remains in the Discussion section mainly because it addresses a question related to that raised by reviewer 1, and, while it does not completely answer the question of the role of the precise FcR subtype or the role of crosslinking in the difference between endocytosis and phagocytosis in BMDCs, the issue was raised in our own minds as a discussion point.

Whether to use 'dislocation' or 'translocation' is something we struggled with. Both have been used in the literature. In the introduction we use 'translocation' to refer to the ER to cytosol transfer that occurs during ERAD. It seemed appropriate to use a different term for antigen transfer from the endocytic/phagocytic pathway and we decided on 'dislocation'. I trust the reviewer has no serious objection to this.

2) Figure 1. Figure 1 c should show bright field images to illustrate that cells are present, even when no luminescence is observed. The luminescence could also be pseudocolored by intensity to better show the differences in signal. Given the lengthy treatment with inhibitors, the viability of the cells should also be reported.

The figure is now updated with the bright field image. We unfortunately cannot perform the pseudocolor analysis requested because we no longer have access to the equipment or appropriate programs. The luminescence microscope was kindly loaned to us for testing by the Olympus company but because of the cost and the overall lack of sensitivity we did not purchase it. It adequately allowed us to see that the luminescence is indeed generated within the cells, both in Fig. 1c and Fig. 2c, and that is all we claim.

To the second point, we have added data on viability that was actually collected during the 293T kinetic experiment, shown in Fig. 3a and b, as Supplementary Figure 6a and b. Up to 8 hours the viability is excellent, and after that time only the Hsp90 inhibitor radicicol has a serious adverse effect compared to the control cells. In the experiments with DCs, viabilities were determined at the termination of the incubations and no issues arose. Data combined from three independent experiments for BMDCs and human DCs are shown below.

3) Figure 2/3. The authors use DMSO as a control, and I presume that the assumption is that DMSO has no effect (it's not clear how much is used and it is not clear how much is present in Epox or sVAD stocks). Untreated cells should be included for comparison (untreated cells and DMSO-treated cells could be different). While it is likely that Epox has an inhibitory effect on cytosolic proteasome, absence of an effect on endosomal degradation should ideally be demonstrated. The authors should also provide additional evidence that the activation of the probe, or its release from endosomes, does not occur during cell lysis (the authors add zVAD in the lysis buffer but do not provide evidence that this is effective under these conditions).

In all experiments DMSO is present at less than 0.5%, now declared in the Methods section under 'The dislocation assay', and no effects on either total cytosolic luciferase entry or deglycosylation-dependent luciferase activity were observed. This is shown below.

Epoxomicin is widely regarded as a very specific proteasome inhibitor. It was checked against many proteases, including cathepsins, in the original papers from Craig Crews that described it. It has also been used in numerous experiments in the antigen processing field to establish proteasome specific effects, so we believe there is extensive data to support its use. It was difficult to come up with an experiment to directly address this issue in the specific context of our data, but the experiment below, for the reviewer but not for publication, shows that phagocytosis of ddRLuc-Fc by 293T-FcR-K^b cells in

the presence of epoxomicin for only one hour does not affect the luciferase activity that is liberated by PNGase F treatment post cell lysis. If early proteolysis of the enzyme precursor in the phagosome was inhibited by epoxomicin, we would expect to see enhancement of activity.

We thank the reviewer for the suggestion regarding the addition of zVAD during lysis. We did this to provide a safety net to ensure that post lysis activation of the enzyme by released NGLY1 did not occur, as the reviewer indicates. To address this issue we have now performed an experiment comparing the results with and without zVAD addition during lysis. It turns out that this precaution was unnecessary. The results were identical in each case, as shown below. We have not included this in the manuscript but the data are provided for the reviewer.

4. The cells treated with the combinations of inhibitors should be tested for viability and proliferation. This is because apoptosis can mediate the permeabilization of membranes. In particular, a concern with this assay may be that a few cells (dying cells) could cause the release and activation of a lot of probe. Figure 2c addresses this concern in part. However, it is only qualitative and does not rule out the possibility that a few super luminescent cells contribute to the majority of the signal detected in the lysate.

While we did not address the issue of proliferation, which we would not expect to be extensive during the periods of study, we did assess viability, as described above and now shown for 293T-FcRγIIa-K^b cells in Supplementary Fig. 6 a and b. In terms of release of the NGLY1 enzyme and activation of the probe caused by the death of a few cells I would point out that in solution the half-life of the glycosylated probe at 37 °C is only 20 minutes (Fig. 1g). I believe its ready detection as an active enzyme 24 hours post phagocytosis and endocytosis (Fig. 3) excludes this possibility. We also include experiments involving cytochalasin D and Dynasore treatment to ensure luminescence was not generated from residual un-internalized probe that could be potentially activated by NGLY1 released from dead cells.

The issue of a few luminescent cells making a major contribution to the total activity is difficult to exclude. We added the bright field image to Fig. 2c as we did for Fig. 1c, but in this case the presence of excess latex beads obscures cells that might not be luminescent, and unfortunately the luminescence microscope was too insensitive to allow this experiment with endocytosed soluble substrate. However, we would point out the excellent correlation between luminescence in the presence of epoxomicin and cross-presentation in its absence, seen for both BMDCs (Fig. 4e and f) and 293T-FcRγIIa-K^b cells (Fig. 5b and c). For the suggestion of the reviewer to be true, this would mean that experiments assaying cross-presentation would face the same issue, including all such experiments in the literature. On this basis we do not think this is likely to be a serious problem.

6) The kinetics in Figure 3b are significantly different than 3a, in particular in how it responds to inhibitors. why is the baseline at 20% even when there is no protein? Cell viability/stress after long term exposure to inhibitors?

The problem with the high baseline was mainly because the signal generated by endocytosis (Fig. 3b) was significantly lower than that with phagocytosis (Fig. 3a). We have removed this perception by subtracting machine background from both, which improves the figure. Nevertheless, some differences do remain. For example, a substantial amount of activity in the case of phagocytosis is epoxomicin independent. This may mean that substrate introduction into the cytosol from phagosomes results in more rapid cytoplasmic folding before proteasomal degradation can occur. The activity is still inhibited by radicicol, however, consistent with a requirement for refolding. Once again, these are interesting mechanistic questions beyond the scope of this manuscript. The issue of viability was addressed above.

3) Supp info 3a.: the authors test the purity of the cytosolic fraction by showing exclusion of lysosomal proteins. This approach should be extended to other endocytic organelles: early endosomes, late endosomes, multivesicular bodies, recycling endosomes. This is necessary to show that the probe release after the treatment with SLO does not coe from endosomally trapped material.

This is difficult because any markers that we use in these experiments, which use cytosol released by SLO, must be both soluble and luminal. Membrane associated proteins commonly used as markers will not be released, and cytosolic markers such as Rab proteins cannot be used. However, we would note that one of the markers we used was prosaposin, which is actually processed in lysosomes or phagolysosomes to mature saposins, which are small molecules, and thus its detection in the pro-form

(the antibody is specific for the pro form) means it is likely coming from early phagosomes in the experiment shown. Thus, the fact that neither prosaposin nor cathepsin D are found in the cytosolic fraction reassures us that the probe released does not come from within the phagosomal pathway. In addition, the cytosolic enzyme NGLY1 is required to generate luciferase activity after endocytosis or phagocytosis, making it even less likely.

4) "this strongly supports a role for VCP/p97...": this conclusion is unclear and requires a comparison between Fig2 and Supp Fig 4...see comment 1.

We apologize for the requirement for toggling between the manuscript figures and supplementary figures, but unfortunately it is difficult to avoid. Our conclusion based on this and earlier data is that VCP/p97 plays a role in the dislocation event. This view is strengthened by the new experiment described at the beginning of this letter and shown in Supplementary Figure 7, where the addition of the Hsp90 inhibitor after completion of cytosolic entry increases the rate of decay of luciferase activity, while the addition of the VCP/p97 inhibitor does not (discussed on p. 5 at the end of the second complete paragraph).

5) Figure 4e. Is it possible that phagosomes/endosomes containing beads are more easily disrupted during lysis than regular endosomes?

The assays are conducted after lysis of the cells using a specific reagent that minimizes background luciferase activity and stabilizes it. The manufacturers assure us it involves a detergent of some kind, although they refuse to divulge exactly what it is. However, we have no reason to believe that phagosomal or endosomal membranes would behave differently from each other or any other membrane.

6) several probes have been reported in the CPP field to detect cytosolic penetration. The authors should mention some of this work in their intro and in their discussion.

We thank the reviewer for pointing out this important issue. We now mention it and add references to the 5th and 6th lines of the introduction.

Reviewer #3 (Remarks to the Author):

This manuscript reports the development and use of a recombinant protein that can be used to detect and quantify the translocation of proteins from vesicular compartments into the cytosol of cells. The approach is really ingenious and elegant. Assays using this probe have an excellent signal-to-noise ratio. The authors do an excellent job of establishing the validity of their assay. Current reagent/approaches to measure vesicle-to-cytosol translocation are not very good, so the author's system is a major technical advance that will be useful to the field. The authors use their probe to quantify and characterize the

translocation in various cells and in cross presentation with interesting results. They show that more cells can translocate internalized proteins, albeit with very different efficiencies, than were generally thought and document differences in this process in different dendritic cells. They give insight into the kinetics of this translocation. They confirm the role of HSP90 and p97 in this process giving insight into mechanism. Another important finding is a remarkable correlation between dislocation and antigen presentation suggesting the antigen transfer is a rate limiting in cross presentation.

There are some minor points that the authors may want to consider but are not essential to address.

1. The authors quantify the number of molecules of probe that are translocated into the cytosol. It would be of interest to quantify the amount of probe that was internalized and calculate the percent translocated.

We agree that this would be of great interest, but it is difficult to address experimentally. The major problem is the instability of the free glycosylated ddRLuc-Fc compared to its relative stability after translocation, which is apparent when comparing Figs. 1g, 3a and 3b. Determining the exact number of molecules based on the activity that binds to the Fc receptor and internalizes is the biggest obstacle. How long would one wait to provide evidence for complete internalization? One might be able to perform calculations that attempt to answer the question but I doubt the reviewer would trust the results.

2. In Fig 3, many of the groups were incubated with proteasome inhibitors for long periods (up to 24 hours) and with many cell types (I don't know about 293t) this will lead to a loss of viability. While this is probably not a problem at the early time points (which are the most informative ones), it would be useful to know whether viability was affected over the time course of the experiment? Any effect of derangement of cell function might be assessed by pre-treating with proteasome inhibitors for various times before adding the probe and seeing if the dislocation kinetics are altered (provided the proteasome inhibitor doesn't reduce the internalization of the probe).

Viability issues were addressed as described above and the results are now included in the manuscript. We have not done the experiment suggested of adding the proteasome inhibitor for different periods prior to initiating internalization of the probe, although we agree that it would be of interest to determine the effects of epoxomicin on both internalization and dislocation. Differentiating between the two might be problematic but we will bear this in mind for future experiments

3. It would be of interest to compare the kinetics of dislocation (Fig 3) with the kinetics of cross-presentation.

It would indeed be interesting. However, cross-presentation by its nature requires that we not add epoxomicin to the cells, whereas dislocation is *revealed* by adding it. Thus the conditions are quite different. It could be useful to look at phagocytosis in the absence of epoxomicin in this context, given

that a significant amount of luciferase activity is detected under these conditions (Fig. 3a). However, elements other than proteolysis are involved in the generation of K^b-SIINFEKL complexes and their survival, including TAP-mediated peptide transport and turnover at the cell surface. Interpretation would be quite complicated and we prefer to leave that issue for future experiments.

4. A casual reader may look at graphs, e.g. Fig 2a and 2b, and interpret them to show an inhibition by DMSO. The authors might want to consider moving this group to be the first bar and labeling it "none" or "control" on the graph, and also somehow showing that the statistically significant change is an increase in the epox group (not a decrease from DMSO); e.g. separate horizontal bars over the DMSO vs Epox and Epox-vs other groups. The authors might want to include data somewhere comparing DMSO to just media, which I presume are not different.

Reviewers' comments:

Reviewer #1 (Remarks to the Author):

The authors addressed all my concerns.

Reviewer #2 (Remarks to the Author):

The authors made some significant changes in their revised manuscript and they have addressed several of the previous comments. Yet, in my view, an important issue remains unclear. This has to do with the problem of few cells possibly giving rise to a large part of signal. This is important because this can change how the results are interpreted. It is especially vital because any future study using this probe would use this study as a foundation: the foundation should therefore be as solid as possible.

The new images provided in Figure 2c certainly suggest that only a few cells (<5%??) show high luminescence signal (although it is hard to tell without a pseudocolored scale of intensity). It seems critical to address whether these cells are healthy. New data are shown about cell viability. Yet, while the authors report no statistical difference, it appears that some of the epox and epox+zVAD may have an effect. It is unclear how the authors measure viability (is vitality really 100% in untreated or are the data simply normalized), making it hard to assess whether these differences are indeed meaningful or not. More importantly, it is the cells that show high luminescence that should be directly tested for viability (by combining luminescence imaging with fluorescence imaging of viability probes?...alternatively, cell death could be induced to test whether this impacts the probe signal?): 2% of cells in the process of dying may not seem like much in terms of loss of viability, but they could contribute dramatically to the signal of the probe.

The authors point out to the correlation between cross-presentation assays and luminescence assays to support the idea that they don't anticipate a problem with the few cell/high signal issue. It is however possible that the cross-presentation assay suffers from the same problem in this particular instance (it doesn't mean that it is a problem in all experiments reported in the literature). Chloroquine is quite toxic, so the correlation between dislocation and cross-presentation could have to do with it.

Reviewer #3 (Remarks to the Author):

None

Reviewer #2 additional comments:

The authors made some significant changes in their revised manuscript and they have addressed several of the previous comments. Yet, in my view, an important issue remains unclear. This has to do with the problem of few cells possibly giving rise to a large part of signal. This is important because this can change how the results are interpreted. It is especially vital because any future study using this probe would use this study as a foundation: the foundation should therefore be as solid as possible.

As pointed out in our previous rebuttal letter it is not trivial to definitively address the issue of whether a few cells might give rise to a major proportion of the signal. As we also pointed out this may also be true for cross-presentation, not just for dislocation into the cytosol measured using our reagent. Luminescence generation depends on many factors, including antigen internalization, phagosomal degradation, proteasomal degradation, etc., and cells within a population may behave differently. However, given this reviewer's concern we contacted the Olympus company who loaned us the luminescent microscope and asked if the files we generated could be used for quantitation. They suggested the use of the program Image-J and we applied this to the relevant figures in the manuscript (Figs. 1C, showing cells transfected with the ddRluc vector, and measuring ERAD, and 2C, showing cells that had phagocytosed latex beads coated with the ddRluc-Fc reagent). We switched to a gray scale rather than the original red, which better represents the intensity range, and the figures now include a brightness scale bar covering a range of zero to 255. In both figures there is a range of brightness in the presence of epoxlmycin, but in neither case can it be said that a few cells give rise to a majority of the signal. In Fig. 2C we calculate that the four brightest cells that are visible (indicated below), which represent approximately 10% of the total luminescent cells in the image, together contribute about 25% of the total signal (Fig. 2c, Epox). These numbers are too small for a complete statistical analysis but do not indicate that a small number of cells are responsible for all, or even the majority, of the signal. Unfortunately we no longer have the microscope and are unable to expand the study to obtain better statistics, and we therefore have not included this quantitation in the manuscript.

It seems critical to address whether these cells are healthy. New data are shown about cell viability. Yet, while the authors report no statistical difference, it appears that some of the epox and epox+zVAD may have an effect. It is unclear how the authors measure viability (is vitality really 100% in untreated or are the data simply normalized), making it hard to assess whether these differences are indeed meaningful or not. More importantly, it is the cells that show high luminescence that should be directly tested for viability (by combining luminescence imaging with fluorescence imaging of viability (by combining

luminescence imaging with fluorescence imaging of viability probes?...alternatively, cell death could be induced to test whether this impacts the probe signal?): 2% of cells in the process of dying may not seem like much in terms of loss of viability, but they could contribute dramatically to the signal of the probe.

The viability of 293T-FcR-K^b cells during the time course study shown in Supplementary Fig. 6a-b was determined by Trypan blue staining, as stated in the legend, and was not normalized. The viability of BMDC (3 h) and hDC (6 h) we showed previously was also determined by Trypan blue staining but in these cases it was normalized to that of the untreated cells. The non-normalized data is shown below and does not affect the outcome. Normalization helped the statistical analysis because there was some variation in viability between experiments, independent of treatment.

We are unable to combine luminescence imaging and fluorescence imaging because we do not have access to an appropriate instrument. However, we have used alternative approaches to investigate this issue. Because active, deglycosylated, ddRLuc-Fc in the cytosol is soluble (Supplemental Fig. 3), we did not think it likely that dead cells could be responsible for the luminescence because the cytosol would be released upon cell death. The luminescence associated with the cells in Fig. 2C clearly maintains cellular morphology. Nevertheless, to directly address this issue we sorted 293T-FcR γ IIA-K^b cells that had phagocytosed ddRLucFc (for 8 hours in the absence and presence of the various drugs) based on GFP expression (the FcR γ IIA construct incorporates a GFP tag) and PI staining. Three populations: GFP-negative cells, which include a small number of non-FcR expressing cells plus cell debris), GFP-positive, PI-positive cells, which are dead/apoptotic, and GFP-positive, PI-negative cells, which are live; were isolated and assessed for luciferase activity. Representative cell sorting data of epoxomycin-treated samples is shown below (left), and the results of luciferase assays (mean of two independent experiments) are in the right panel.

Epoxomycin enhanced the signal, as expected, and this was blocked by zVad. However, approximately 90% of the total signal, both in the presence of DMSO only (the control) and in the presence of epoxomycin, was attributable to live cells. Furthermore, as the reviewer suggested, we induced cell death deliberately, using 32 μ M Actinomycin D (Act D). This reduced the total signal by almost 80%, further confirming that the majority of the luminescence derives from healthy cells.

The authors point out to the correlation between cross-presentation assays and luminescence assays to support the idea that they don't anticipate a problem with the few cell/high signal issue. It is however possible that the cross-presentation assay suffers from the same problem in this particular instance (it doesn't mean that it is a problem in all experiments reported in the literature). Chloroquine is quite toxic, so the correlation between dislocation and cross-presentation could have to do with it.

We agree with the reviewer that not all experiments on cross-presentation may result from a few cells in a population being extremely active. However, this has rarely been assessed, for example by FACS analysis using a mAb that is specific for particular MHC-I-peptide complex. We attempted to stain our cell populations using the K^b-SIINFEKL-specific 25.D1 mAb but the signal was insufficient. We also examined the viability after 8 hours of phagocytosis of chloroquine-treated 293T-FcRγIIA-K^b cells (50 μM) by FACS and PI staining, calculating viability as the percentage of GFP-positive, PI-negative cells out of the total GFP-positive cells. The figure below shows the mean result of two independent experiments. The inclusion of chloroquine with epoxomycin had no detectable effect on viability. Clearly, given these data and the assay results following cell sorting shown above, healthy cells are responsible for the great majority of the signal in all our experiments.

We believe that these data should satisfy the reviewer's remaining concerns.

REVIEWERS' COMMENTS:

Reviewer #2 (Remarks to the Author):

the authors have addressed my remaining concerns.